# Scaling Laws for Upcycling Mixture-of-Experts Language Models

**Seng Pei Liew** [1]   **Takuya Kato** [1]   **Sho Takase** [1]

## Abstract

Pretraining large language models (LLMs) is resource-intensive, often requiring months of training time even with high-end GPU clusters. There are two approaches of mitigating such computational demands: reusing smaller models to train larger ones (upcycling), and training computationally efficient models like mixture-of-experts (MoE). In this paper, we study the upcycling of LLMs to MoE models, of which the scaling behavior remains underexplored. Through extensive experiments, we identify empirical scaling laws that describe how performance depends on dataset size and model configuration. Particularly, we show that, while scaling these factors improves performance, there is a novel interaction term between the dense and upcycled training dataset that limits the efficiency of upcycling at large computational budgets. Based on these findings, we provide guidance to scale upcycling, and establish conditions under which upcycling outperforms from-scratch trainings within budget constraints.

## 1. Introduction

Large-scale neural network architectures, such as dense transformers (Vaswani et al., 2017), have seen remarkable success across a wide range of tasks, particularly achieving human-level capabilities in natural language processing (Achiam et al., 2023). However, they often demand an enormous amount of computation, imposing challenges of computational efficiency and scalability. Sparse models like mixture-of-experts (MoE) architectures (Shazeer et al., 2017; Lepikhin et al.) have emerged as an alternative achieving better efficiency-performance trade-off via partial activation (routing) of neural parameters when processing the input. Even so, MoE models still require substantial compute power to reach its full potential (Wei et al., 2024;

[1]SB Intuitions, Tokyo, Japan. Correspondence to: Seng Pei Liew <sengpei.liew@sbintuitions.co.jp>.

*Proceedings of the 42$^{nd}$ International Conference on Machine Learning*, Vancouver, Canada. PMLR 267, 2025. Copyright 2025 by the author(s).

Dai et al., 2024; Yang et al., 2024).

One direction to further accelerate training convergence is leveraging smaller pretrained models to guide the training of larger MoE models. Komatsuzaki et al. proposed upcycling, which reuses the dense checkpoint to continued pretrain the upcycled MoE. The MoE is expected to specialize and optimize routing more rapidly by leveraging the pretrained dense model weights.

Despite the promise of efficient MoE training via upcycling, the effectiveness and limitations of this technique remain unclear. While some (Wei et al., 2024; He et al., 2024) have already adopted it to training large-scale MoE models, Muennighoff et al. (2024) reported negative results where upcycling can slow down training convergence. We believe these seemingly contradictory conclusions are due to insufficient comprehensive studies and assessments. There is also a lack of guidance on how and when to upcycle, hampering a wider adoption of this technique.

In this paper, we seek to better understand large language models' (LLMs) upcycling to MoE models via a series of controlled experiments, spanning up to a few hundred billion (B) training tokens and models up to 7B total parameters. Specifically, we uncover precise power-law scalings for the language modeling performance (cross-entropy loss) with respect to training dataset size (for both dense and upcycled MoE training), and the model configuration, including the total number of parameters (model size). Building on these results, we provide a framework for assessing when upcycling offers advantages over from-scratch training and how performance gains depend on dataset size and model configuration.

**Main results.** The major technical findings of this paper are summarized below. Let $D_1$, $D_2$ be the number of tokens used to train the dense model and upcycled MoE respectively. Denote the cross-entropy test loss of the upcycled MoE by $L$. We find that the upcycled MoE satisfies the following relation for a wide range of model configuration:

$$L = AD_1^{-\alpha_1} D_2^{-\alpha_2 + \alpha_3 \log D_1} + E \qquad (1)$$

where $\alpha_i$'s ($i = 1, 2, 3$) are positive scaling exponents, and $A, E$ are constants independent of $D_1, D_2$.

Moreover, the empirical performance has a power-law scal-

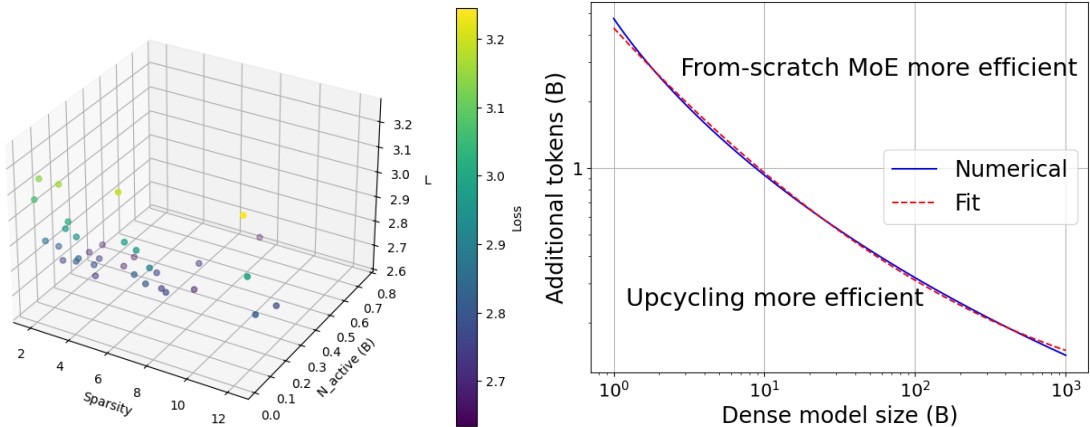

*Figure 1.* **Left: Upcycling improves with sparsity and the number of active parameters.** We find that upcycling to MoE which is sparser and has more active parameters improves performance. The z-axis shows the value of cross-entropy loss. See Section 4.2 for details. **Right: Efficiency of upcycling diminishes with sunk cost and model size.** For all additional token budgets for upcycling a Mixtral-like MoE above the curve(s), training MoE from scratch is more efficient, whereas for all token budgets below it, upcycling is more efficient. Shown are the numerical (blue) and analytical (red) solutions of Equation 13. See Section 5.1 for details.

ing with respect to the model size as well as its sparsity (Equation 11). These allow us to quantify the effectiveness and shortcomings of upcycling. Particularly, our empirical results suggest that :

- Increasing $D_1$ (sunk cost) reduces the initial losses of the upcycled MoE but results in slower training progress with $D_2$ (upcycled MoE tokens).

- Upcycled MoE benefits from increased sparsity and active parameters without noticeable trade-offs. See the left panel in Figure 1.

- We propose a joint scaling law of dataset and model sizes for Mixtral-like MoE (Jiang et al., 2024) (Equation 12). While we find that the advantage of upcycling diminishes with increasing sunk cost and dense model size ($N_1$), upcycling remains effective when $D_2$ are limited.

- Particularly, upcycling is beneficial when $D_2$ remains below a certain threshold, where the pretrained dense model can still accelerate convergence. However, beyond this threshold, from-scratch training of MoE becomes more efficient. The threshold $D^*$ is

$$D^* \simeq 4 \left( \frac{N_1}{10^9} \right)^{-0.7+0.04 \log(N_1/10^9)} \text{ B tokens} \quad (2)$$

See the right panel of Figure 1.

**Notations.** We summarize main notations used in the paper.

- $L$: cross-entropy loss in nat

- $D$: dataset size in token
- $N$: non-embedding model size
- $A, B, F$: scaling factors of the power law, independent of the variable under consideration
- $E$: irreducible loss of the power law, independent of the variable under consideration
- $\alpha, \beta, \gamma$: scaling exponents of the power law, independent of the variable under consideration

## 2. Preliminaries

### 2.1. Model details

**Dense model.** Our dense models are decoder-only transformers pretrained with an autoregressive language modeling objective. The architecture is most similar to Llama2 models (Touvron et al., 2023), incorporating advances such as SwiGLU (Shazeer, 2020) and rotary position embedding (Su et al., 2024). We use the Llama tokenizer of vocabulary size 32,000.

**Mixture-of-Experts.** The MoE in consideration is the same as our dense model, but with all MLP blocks replaced by multiple blocks (experts) with the same configuration (Fedus et al., 2022). A router consisting of a single-layer MLP outputs the routing probability of the tokens to the experts.

The model configuration has two key parameters: $n_{\text{expert}}$, representing the number of experts, and $n_{\text{TopK}}$, which specifies how many of the highest-probability experts each token is routed to at each layer. The output of the experts is linearly combined and passed to the next layer.

The MoE and its corresponding dense model with model size $N_{\text{dense}}$, consisting of $n_{\text{expert}}$ experts, is denoted with

a prefix "$n_{\mathrm{expert}}$", e.g., 8x1B where the dense correspondent is of 1B in model size. We refer to the number of non-embedding model parameters (that is, total number of parameters minus the number of embedding and language modeling head parameters) used for computation per token as the number of *active parameters*.

**Upcycling.** The upcycling scenario assumes that one is given a pretrained dense model and would like to train an MoE with the same configuration but replacing the MLPs with the MoE modules (Komatsuzaki et al.). By replicating the dense model's MLP weights $n_{\mathrm{expert}}$ times to form the experts, the knowledge from the dense model can be reused to accelerate the training of the MoE compared to training the MoE from scratch (from random parameter initialization). Other modules' weights are copied from the dense counterparts directly, with the router's initial weights randomized. See Figure 2 for an illustration of upcycling. We employ this technique for our study. See Appendix A.6 for other details, including alternative upcycling methods.

## 2.2. Power-law Ansatz

There is an extensive literature showing that the loss of training deep learning models has a simple power-law behavior: $L = \frac{A}{X^\alpha} + E$, for single variable $X$, including model size and dataset size (Hestness et al., 2017; 2019; Rosenfeld et al.; Henighan et al., 2020)[1]. We use this ansatz in this work.

Hoffmann et al. (2022) have shown that when training a dense transformer, the cross-entropy loss is well-described by the following "Chinchilla" scaling law:

$$L = AD^{-\alpha} + BN^{-\beta} + E \qquad (3)$$

The first and second terms quantify the limitation of learning due to limited dataset and model sizes respectively. The scaling exponents, $\alpha$, $\beta$ control how fast the loss decreases with respect to dataset and model sizes respectively. $E$ is a constant: it is the irreducible loss representing the inherent entropy of text. All parameters are to be fitted with experimental observations. We also assume Equation 3 for models trained from scratch.

## 3. Experimental Design and Results

In this Section, we describe the design of our experiments, before presenting some of the experimental results. Additional details, including ablation studies are available in Appendix A.

---

[1]The actual form we assume is $L = \frac{A}{(X+1)^\alpha} + E$ such that the loss is finite at the limit $X \to 0$. However, since the values of $X$ we consider are often much larger than 1 ($10^6$ or more), we approximate it as $L = \frac{A}{X^\alpha} + E$ for notational convenience.

### 3.1. Setup

**Dense models in consideration.** We train a suite of dense models with model sizes 15 million (M), 44M, 0.1B, 0.2B, 0.5B and 1B. The model configuration (number of layers, $n_{\mathrm{layer}}$, hidden dimension $d_{\mathrm{model}}$, and MLP hidden dimension, $d_{\mathrm{mlp}}$) used in this paper is summarized in Table 3 of Appendix A.

We denote $N_1$ by the dense model's total number of non-embedding parameters, $N_2$ by the MoE's total number of non-embedding *active* parameters, $N_{\mathrm{total}}$ by the MoE's total number of non-embedding parameters. They are given by (ignoring subleading contribution proportional to $d_{\mathrm{model}}$)

$$N_1 \approx (4 + 12)n_{\mathrm{layer}}d_{\mathrm{model}}^2 = 16n_{\mathrm{layer}}d_{\mathrm{model}}^2 \qquad (4)$$
$$N_2 \approx (4 + 12n_{\mathrm{TopK}})n_{\mathrm{layer}}d_{\mathrm{model}}^2 \qquad (5)$$
$$N_{\mathrm{total}} \approx (4 + 12n_{\mathrm{expert}})n_{\mathrm{layer}}d_{\mathrm{model}}^2 \qquad (6)$$

where the attention and MLP module contribute $4d_{\mathrm{model}}^2$ and $12d_{\mathrm{model}}^2$ parameters respectively per layer.

**Learning rate schedule.** We are interested in training models with different numbers of training token budget. Previous work (Hoffmann et al., 2022) used the cosine learning rate (LR) schedule and trained separate models for *each* number of training token budget, which is resource-consuming. We instead employ the warmup-stable-decay (WSD) learning rate schedule (Bi et al., 2024; Hu et al., 2024), which requires only a single model sweep with a sufficiently large number of training tokens. As the LR is constant for the majority of learning, the saved intermediate model checkpoints can be reused to emulate different numbers of training token budget; the checkpoints are continued-pretrained reusing the optimizer states from the constant LR stage with a new, shorter LR schedule. This reduces substantially the number of new training runs required. In Appendix A.4, we show that the performance of dense and MoE training is on par with the usual cosine LR schedule. The maximum LR value is tuned individually for each model size and for both dense and MoE training.

**Dataset.** We use training dataset derived from the CommonCrawl portion of Slimpajama-DC (Shen et al., 2023), containing 368B tokens in total. The test loss is calculated from the default validation set (0.3B tokens) defined therein. In Appendix B, we train models on two different datasets (Japanese language and source code datasets) to show that the scaling behavior generalizes across datasets.

### 3.2. Upcycled Training

Upcycled training is performed by first initializing the MoE with the dense pretrained models as mentioned before. Subsequently, we train the upcycled MoE with standard cross-entropy loss augmented by an auxiliary load-balancing loss with coefficient $10^{-3}$ to minimize expert's imbalance of

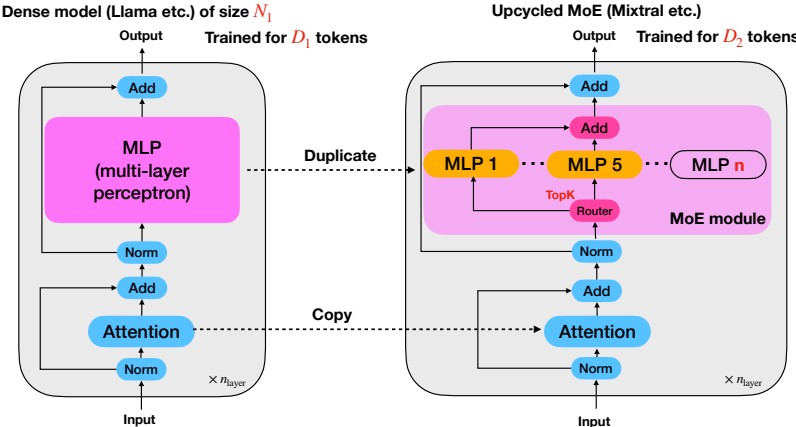

*Figure 2.* **Upcycling and factors affecting MoE's performance.** Upcycling involves initiating the weights of the MoE (activating $n_{\text{TopK}}$ experts per token) by reusing the weights (duplicating the weights of MLPs $n_{\text{expert}}$ times) of an existing dense transformer of size $N_1$ that has been trained for $D_1$ tokens. The (upcycled) MoE is further trained for $D_2$ tokens. Language modeling performance improves when scaling these factors. We study and develop formulae (scaling laws) consisting of these factors to predict the empirical performance.

activation. See Appendix A.7 for ablation of the coefficient.

To this end, we obtained three sets of results in total, of which we denote by: **dense training** (dense model training from scratch), **MoE training** (MoE training from scratch), **upcycled training** (MoE training for $D_2$ tokens where the dense model pretrained for $D_1$ tokens has been reused).

### 3.3. Training results

We show some of the resulting test loss curves of our upcycling experiments in Figure 3 (see model evaluation with standard benchmarks in Appendix A.9). Some observations can be made from the plot:

(a) upcycling from a dense model with more pretrained tokens leads to lower initial losses.
(b) The more overtrained a dense model is, the smaller the rate of the final loss change of the upcycled MoE becomes (the exponent $\alpha$ becomes smaller).

The reason for (a) is *function preservation*. As all experts inherit the same weights from the dense model, the output of the MoE module is preserved from the dense model irrespective of the routing at initialization, and therefore the loss is preserved as well. For (b), our intuitive explanation is as follows. The duplicated experts' weights are already close to the optimal ones when they are upcycled from an overtrained dense model. It is then harder for the experts to diversify and specialize at the MoE training stage to further lower the loss. As we will reveal soon, our proposed scaling law captures these phenomena.

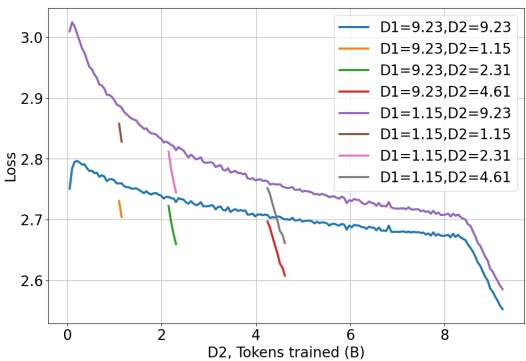

*Figure 3.* **Loss curves of upcycling.** Intermediate test losses of the 8x0.1B MoE (2 experts activated per token) trained for a variety of total number of tokens, $D_2$, when upcycled from a dense model pretrained with various numbers of training tokens ($D_1$) in B.

## 4. Scaling Laws

Our ultimate goal is to understand the performance of upcycling with respect to various factors illustrated in Figure 2. However, simultaneously varying all these variables to investigate the scaling behavior is computationally prohibitive. We separately investigate two aspects closely related to upcycling: dataset sizes (Section 4.1) and model configuration (Section 4.2). In the next Section, we further study the joint scaling behavior of dataset and model sizes with a predetermined MoE architecture.

### 4.1. Scaling Law for Dataset Sizes

We fix the model size while varying $D_{1,2}$ to study the scaling behavior with respect to these variables. To determine the functional form of the scaling law, $L(D_1, D_2)$, we require

it to satisfy certain properties:

**Requirement 1.** $L(D_1, D_2)$ follows the power law with respect to $D_2$, as shown empirically in Figure 4. This aligns with the power-law ansatz, treating upcycling as analogous to standard MoE training with dataset size $D_2$, initialized with dense parameters rather than random weights:

$$L(D_1, D_2) = L_{D_1}(D_2) = AD_2^{-\alpha} + E \qquad (7)$$

**Requirement 2.** As $D_2 \to 0$, the loss should reduce to the power-law scaling behavior of the dense counterpart with respect to $D_1$, consistent with the function-preserving initialization of upcycling (see Section 3.3):

$$\lim_{D_2 \to 0} L(D_1, D_2) = AD_1^{-\alpha} + E$$

We investigate the following functional forms of power law satisfying these requirements:

**Multiplicative:**

$$L(D_1, D_2) = AD_1^{-\alpha_1}(1 + D_2)^{-\alpha_2 + \alpha_3 \log D_1} + E$$
$$\approx AD_1^{-\alpha_1} D_2^{-\alpha_2 + \alpha_3 \log D_1} + E \qquad (8)$$

**Additive:**

$$L(D_1, D_2) = AD_1^{-\alpha_1} + F(1 + D_2)^{-\alpha_2 + \alpha_3 \log D_1} + E$$
$$\approx AD_1^{-\alpha_1} + FD_2^{-\alpha_2 + \alpha_3 \log D_1} + E \qquad (9)$$

Both forms include an *interaction* term ($\alpha_3 \log D_1$) in the scaling exponent, capturing the interplay between $D_1$ and $D_2$.

**Empirical comparisons of functional forms.** We empirically compare these functional forms, including the special case where $D_1$ and $D_2$ have no interaction. The optimization uses the Huber loss ($\delta = 10^{-3}$) and the BFGS algorithm, fitting the logarithm of the loss via the LogSumExp trick applied to the RHS of Equations 8 and 9. The leave-one-out root mean square error (RMS) serves as the fit metric.

The fit is performed on a 0.1B dense model upcycled to MoE architectures with $n_{\text{expert}} = \{4, 8\}$ and $n_{\text{TopK}} = \{1, 2\}$, trained on a $5 \times 5$ grid of $D_1, D_2$. The fitting results are shown in Table 1, where we can see that the multiplicative functional form (with non-zero interaction) achieves consistently the lowest leave-one-out RMS error across the experimented MoE architectures. Henceforth, we adopt Equation 8 in our scaling laws.

**Multiplicative scaling law from empirical observations.** To strengthen our proposal for Equation 8, we demonstrate through a bottom-up approach that the multiplicative nature of the scaling law, including the interaction term, arises naturally from empirical data.

From Figure 4, we observe that the scaling exponent in Equation 7 decreases as $D_1$ increases. To quantify this relationship, we model the scaling exponent as a function of $D_1$, denoted as $-\alpha(D_1)$. A scatter plot of $-\alpha(D_1)$ reveals the following logarithmic relationship:

$$-\alpha(D_1) = \gamma \log D_1 + E'. \qquad (10)$$

which fits the data well, as shown in the same Figure. Substituting this expression to Equation 7, we obtain a term of the form $D_2^{-E' + \gamma \log D_1}$, directly supporting the interaction term in the multiplicative scaling law. This shows that the interaction term is not merely an artifact of theoretical derivation but is essential to explain observed empirical trends. Additionally, we show that the multiplicative dependence, $A \propto D_1^{-\alpha_1}$ also arises from empirical observations, as detailed in Appendix C.

#### 4.1.1. INTERPRETATIONS

Several quantitative and qualitative observations can be made from the scaling law of dataset sizes. From our fit (Table 8 in Appendix C), we notice a trend $\alpha_2 \gtrsim \alpha_1 \gg \alpha_3$. This means that while increasing either of $D_1$ and $D_2$ helps improve performance, as upcycled training has a slightly larger exponent ($\alpha_2$), increasing $D_2$ helps train faster.

**Upcycled MoE has a better head start (effective scaling factor is smaller).** Fixing $D_1$, we see that the effective scaling factor for $D_2$ is $AD_1^{-\alpha_1}$. Increasing $D_1$ lowers the effective scaling factor, and hence the loss of upcycling. Indeed, fixing $D_2$, we see that the model performs better with increasing $D_1$ in Figure 3.

**Upcycled MoE trains slower with larger sunk cost (effective scaling exponent is smaller).** Again fixing $D_1$, we see that the effective scaling exponent with respect to $D_2$ is $\alpha_2 - \alpha_3 \log D_1$. This means that the larger the sunk cost ($D_1$) is, the loss decreases more slowly with $D_2$, indicating diminishing returns from increasing $D_2$ at higher $D_1$ values, agreeing with Figure 4's results.

| $(n_{\text{expert}}, n_{\text{TopK}})$ | (4,1) | (4,2) | (8,1) | (8,2) |
|---|---|---|---|---|
| **Mul.** | **0.0111** | **0.0081** | **0.0105** | **0.0031** |
| **Mul. ($\alpha_3 = 0$)** | 0.0169 | 0.0085 | 0.0180 | 0.0095 |
| **Add.** | 0.0165 | 0.00843 | 0.0167 | 0.0093 |
| **Add.($\alpha_3 = 0$)** | 0.0196 | 0.0117 | 0.0430 | 0.0113 |

*Table 1.* **Multiplicative scaling law with interaction consistently achieves lowest error.** Leave-one-out RMS error for fitting the loss for MoEs upcycled from a dense 0.1B model, with functional forms of Equations 8, 9, and specific cases with $\alpha_3 = 0$. The first and second number in the bracket of the first row indicates the MoE architecture's parameter, $n_{\text{expert}}$ and $n_{\text{TopK}}$ respectively.

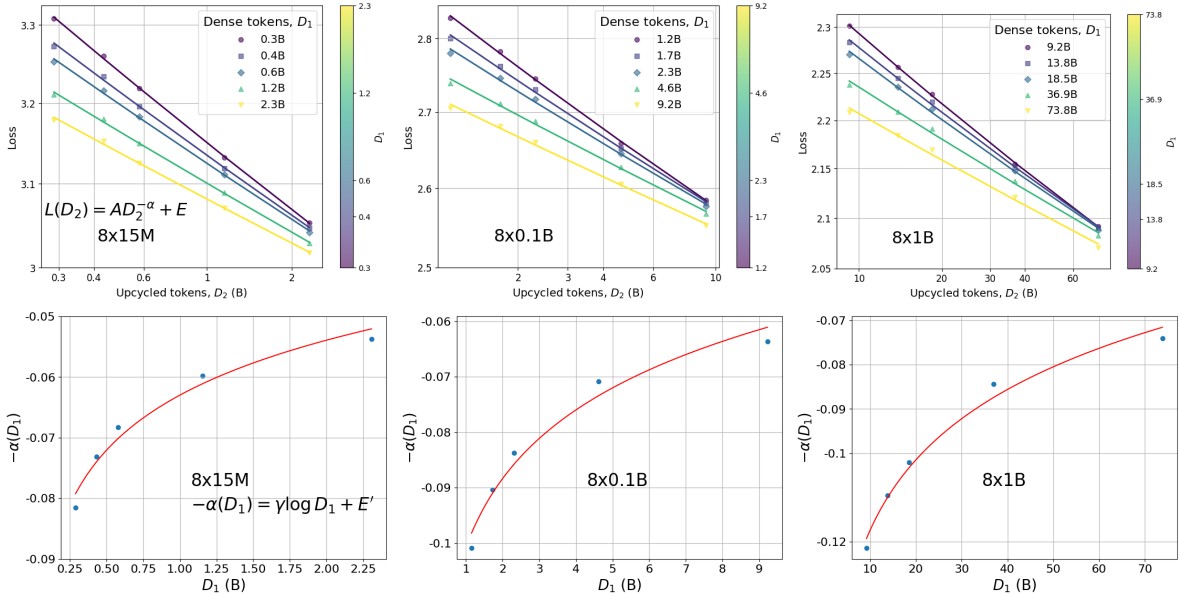

Figure 4. **Top: $D_2$ has power-law scaling.** We show scaling behavior of upcycled training tokens ($D_2$) for different values of dense tokens ($D_1$). **Bottom: Interaction term explains decreasing exponents.** The fitted exponents in the upper plots are used to fit Equation 10 as a function of $D_1$, and are shown to agree well with the functional form.

## 4.2. Scaling Law for Model Configuration

Previous Subsection mainly concerns with *when* to upcycle, i.e., the training tokens used when upcycling. Here, we study *how* to upcycle the dense model, i.e., the model configuration of the MoE.

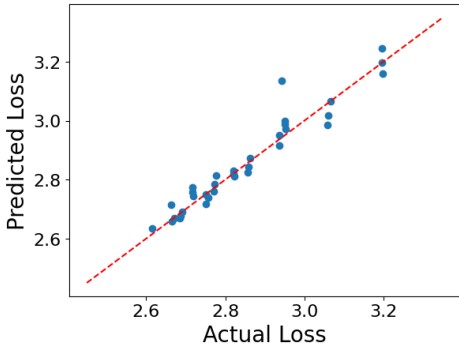

Figure 5. **Fits of scaling law for model configuration.**

To do this, we define *sparsity*, $P := N_{\text{total}}/N_2$, and consider the dependency of the performance on sparsity and active parameter $N_2$, of which the variables capture scaling behavior with respect to model parameters and architectural details ($n_{\text{expert}}$ and $n_{\text{TopK}}$). See Equations 4 to 6.

Using these variables, we can derive and validate the appropriate functional form for the scaling law, similar to the analysis performed in Section 4.1. We fit the functional form by running experiments fixing $D_1 = D_2 = 2.3$B, varying

the following: dense model of size $\{15\text{M}, 44\text{M}, 0.1\text{B}, 0.2\text{B}\}$, $n_{\text{expert}} = \{4, 8, 16, 32\}$ and $n_{\text{TopK}} = \{1, 2, 4\}$. The scatter plot is shown in the left panel of Figure 1 (for better readability, see the 2D slices of the plot with fixed sparsity in Figure 13). To this end, we find that the following functional form provides the best fit to the data:

$$L(P, N_2) = AP^{-\beta_1} + FN_2^{-\beta_2} + E \tag{11}$$

See Appendix D.1 for details.

We obtain the following values for the exponents: $\beta_1 = 1.87$, $\beta_2 = 0.34$. See Figure 5 for the fits. Unlike dataset sizes, due to computational constraints, we are unable to generate more data points to make a more precise estimate extending the range of model configuration. Nevertheless, we do not observe performance trade-offs when scaling sparsity and active parameters within the range of our examination, in contrast to the scaling law of dataset sizes. Our work suggests that: **Performance of upcycling improves with sparsity and active parameters without noticable trade-offs.**

## 5. Joint Scaling Law

In this Section, we show that the upcycled MoE follows a joint scaling law with respect to dataset size and model size, provided the MoE architecture is fixed. While increasing sparsity and active parameters generally improves performance as shown before, compute and memory constraints impose practical upper limits. Thus, while scaling MoE

configurations as much as possible is theoretically beneficial, practitioners should choose an architecture that aligns with their available compute, memory, and expected training token budget. For our study, we adopt a widely used MoE configuration ($n_{\text{expert}} = 8, n_{\text{TopK}} = 2$), which has been implemented in several large-scale, publicly available models, including Mixtral-8x7B and 8x22B (Jiang et al., 2024), and Sarashina2-8x70B [2].

To establish the scaling law with respect to model size, it is sufficient to consider the dependency solely on $N_1$ among other definitions (Equations 4, 5, 6), as they are equivalent up to a multiplicative constant, e.g., $N_2/N_1 \approx 1.75$. To determine the scaling law, we require the functional form to satisfy the following, analogous to previous Requirements:

**Requirement 3.** $L_{D_1,D_2}(N_1) = B_{D_1,D_2}N_1^{-\beta} + E_{D_1,D_2}$.

**Requirement 4.** $\lim_{D_2 \to 0} L(D_1, D_2, N_1) = AD_1^{-\alpha} + BN_1^{-\beta} + E$, i.e., approaching Chinchilla scaling law. The straightforward functional form that satisfies these requirements is as follows [3].

$$L(D_1, D_2, N_1) = AD_1^{-\alpha_1} D_2^{-\alpha_2 + \alpha_3 \log D_1} + BN_1^{-\beta_2} + E \tag{12}$$

**Fitting.** Towards this end, we fit three separate scaling laws, corresponding to dense, MoE, and upcycled trainings, incorporating dataset and model sizes (the first two are fitted with the functional form of Equation 3). We use the same fitting procedure presented in Section 4.2. As the irreducible loss $E$ is universal irrespective of scaling laws under consideration, we first estimate $E$ from a joint fit of dense and MoE scaling laws, then fix it when fitting the upcycling scaling law to reduce overfitting and enhance extrapolation ability.

In Figure 6, we show that the fitted result of upcycling scaling law extrapolates well, achieving validation RMS error of 0.015. The fitted parameters using all data for all three scaling laws are summarized in Table 2. We consider other fitting possibilities and present more results in Appendix E. For the rest of this Section, we consider the application and implication of the joint scaling law.

### 5.1. Training MoE from Scratch versus Upcycling

Past studies (Komatsuzaki et al.; Muennighoff et al., 2024) have explored the efficiency of training an MoE from scratch compared to upcycling a dense model. Komatsuzaki et al.

[2]https://huggingface.co/sbintuitions/sarashina2-8x70b

[3]One could extend this with Equation 11 to create a truly unified scaling law encompassing all model configurations. However, we refrain from doing so in this work, as reliably fitting such a scaling law with many independent variables would require significantly more extensive experimentation beyond our current computational capabilities.

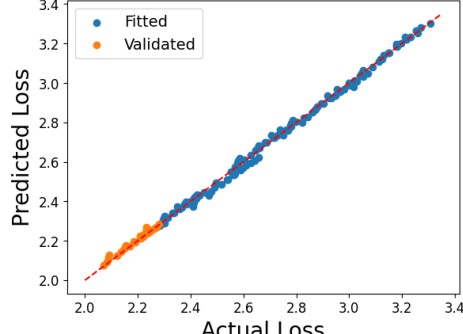

*Figure 6.* **Fits of the joint upcycling scaling law.**

| | $A$ | $B$ | $\alpha/\alpha_1$ | $\alpha_2$ | $\alpha_3$ | $\beta$ |
|---|---|---|---|---|---|---|
| **Dense** | 8.83 | 12.3 | 0.088 | – | – | 0.116 |
| **MoE** | 32.0 | 7.05 | 0.161 | – | – | 0.080 |
| **Upcycled** | 16.3 | 8.53 | 0.043 | 0.085 | 7.98e-4 | 0.112 |

*Table 2.* **Fitted parameters for joint scaling laws.** Note that for MoE, we fit the parameters with variable $N_1$ (instead of $N_2$) for comparison conveniences. The irreducible loss applicable to all laws, $E$, is fitted to be 0.165.

found that upcycling retains an advantage up to 120% of the sunk cost ($D_1$). For instance, to match the performance of an upcycled MoE trained with an additional 0.4 trillion (T) tokens after 2T dense tokens, training an MoE from scratch would require 2.4T tokens—effectively saving 2T tokens in upcycled training. Conversely, Muennighoff et al. (2024) reported that training from scratch requires less than 100% of the sunk cost under different settings, indicating that it could be more efficient. These seemingly contradictory results suggest that upcycling efficiency depends on both sunk cost and model size.

To investigate this, we define $D^*$ the number of tokens required for training from scratch to match the performance of an upcycled MoE with the same sunk cost, i.e.,

$$L_{N_1}^{\text{MoE}}(D^*) = L_{N_1}^{\text{Upc.}}(D_1 = D^*, D_2 = D^*) \tag{13}$$

Since the above equation involves non-integer polynomial exponents, we solve it numerically and approximate the solution analytically (Equation 2). Figure 1 shows that $D^*$ decreases with increasing model size, with $D^*$ equal to 4B tokens for an 8x1B model. When $D_2 \lesssim D^*$, the required $D_{\text{MoE}}$ to catch up is more than 100% of $D_1$: upcycling remains more efficient than training from scratch. However, for $D_2 \gtrsim D^*$, the efficiency reverses, favoring training from scratch. Our findings indicate that: **Upcycling is more efficient than training an MoE from scratch for lower sunk cost and training token budget, but its efficiency diminishes as the sunk cost or model size increases.**

## 5.2. Compute Allocation and Compute-Optimal Upcycling

Given a fixed total floating-point operation (FLOPs) budget $C$, we analyze how to optimally allocate compute between dense training and upcycled training. Specifically, we solve:

$$\min_{D_1, D_2, N_1} L(D_1, D_2, N_1) \text{ s.t. FLOPs}(D_1, D_2, N_1) = C$$

We find that for a given compute budget without any pre-trained models, training a compute-optimal dense or MoE from scratch outperforms the two-stage dense-to-upcycled MoE approach (see Figure 16 in Appendix F.2). This suggests: **If no pretrained model exists, direct training is preferable to upcycling for optimal performance.**

In scenarios where a pretrained dense model is already available, we can determine the compute-optimal scaling of MoE upcycling. The compute cost of upcycled training is approximated as $C_2 = 6N_2 D_2$ (Kaplan et al., 2020). Optimizing the compute leads to the scaling relations (see Appendix F.2 for derivation):

$$D_2^{\text{opt}} \propto C_2^{\frac{\beta}{\beta + \alpha_{\text{eff}}}}, \quad N_1^{\text{opt}} \propto C_2^{\frac{\alpha_{\text{eff}}}{\beta + \alpha_{\text{eff}}}} \tag{14}$$

where $\alpha_{\text{eff}} := \alpha_2 - \alpha_3 \log D_1$. Notably, as $D_1$ increases, $\alpha_{\text{eff}}$ decreases, meaning **larger pretrained models require disproportionately more tokens for efficient upcycling.**

For instance, applying this to the Llama2 models (7B, 13B, 70B), which were trained on 2T tokens (Touvron et al., 2023), we estimate that compute-optimal upcycling follows the scaling $D_2 \propto N_1^{1.8}$, indicating that larger models require nearly quadratic increases in upcycling data.

Overall, we find that upcycling is inefficient relative to from-scratch trainings when considering compute optimality.

## 6. Related Work

**Mixture-of-Experts.** While interests in developing open MoE LLMs are relatively recent (Hu et al., 2024; Yang et al., 2024; Dai et al., 2024; Liu et al., 2024; Sun et al., 2024), the use of MoE in deep learning can be traced to early 2010s (Eigen et al., 2013; Bengio et al., 2013). See Cai et al. (2024) for a detailed survey of modern MoE models.

**Upcycling.** Leveraging pretrained models to expedite the training of larger dense models is also known as model growth (Chen et al., 2015). In the context of training MoEs reusing dense pretrained models, Komatsuzaki et al. are the first studying such a scenario with encode-decoder models. There are studies (Hu et al., 2024; Yang et al., 2024; Lo et al., 2024; Wei et al., 2024; Muennighoff et al., 2024; He et al., 2024) offering insights into upcycling decoder-only transformers. However, a systematic investigation has been lacking. Wei et al. (2024) made only a rough recommendation: use upcycling when the budget for training is smaller than twice the budget used for dense training. In contrast, we have presented a more general guideline.

**Scaling laws.** Power-law scaling appears in a variety of natural and human-made phenomena. Scaling studies that are perhaps closest to our work are those considering two-stage training, e.g., transfer learning, fine-tuning, and model growth (Mikami et al., 2022; Zhang et al.; Du et al., 2024). Prior work on transfer learning has proposed scaling laws using multiplicative or additive forms involving $D_1$ (pretraining data) and $D_2$ (fine-tuning data) (Mikami et al., 2022; Zhang et al.). A notable similar phenomenon in transfer learning is *ossification*, where pretraining can hurt fine-tuning performance (Hernandez et al., 2021). However, to our knowledge, no prior work incorporates an interaction term ($\alpha_3$ in Equation 1) to capture such effects. Although our experiments specifically focus on upcycling into MoE models, we believe that the core insights, such as the interaction between dense and upcycled training budgets, are useful for formulating two-stage training regimes more broadly, including transfer learning as mentioned above, and potentially model growth.

In the context of MoE, Clark et al. (2022) studied how different architectural choices affect MoE's scaling, while Krajewski et al. (2024) investigated fine-grained experts' scaling behavior. The latter work fits a joint scaling law with respect to dataset and model sizes, but we find differences in the obtained parameters. It is likely due to several differences in methodology: the largest dense model they experimented with was smaller (85M), and they did not tune the LR for each model size. Nevertheless, our findings that MoE scales better than its dense counterpart with sufficiently large computational budget do agree with theirs.

## 7. Discussion and Conclusion

In this paper, we have presented compelling evidence that MoE upcycling follows novel scaling laws with dataset and model configuration, revealing trade-offs due to interactions between dataset sizes.

Given the complexity of upcycling, deriving a unified scaling law that allows for simultaneous optimization of dataset size and model configuration is beyond our current computational reach. However, our findings still provide practical guidance, as the joint scaling law (Equation 12) enables accurate predictions of upcycling performance once the MoE architecture is fixed. In practice, we recommend selecting the largest viable MoE size and sparsity within compute and memory limits while balancing this choice against the expected training token budget. Once the architecture is set, Equation 12 can guide dataset/model scaling, demonstrating reliable extrapolation of upcycling performance.

While our empirical formulae successfully capture the ob-

served scaling behavior, the underlying mechanism, particularly the interaction term, remains theoretically unexplained. To our knowledge, formal justification for such a term is lacking in the literature (Paquette et al.). Future work could explore it through, e.g., simplified models (Hutter, 2021; Maloney et al., 2022; Lin et al.; Bordelon et al., 2024; Bahri et al., 2024; Paquette et al.). Additionally, extending our scaling laws to alternative MoE architectures (shared experts and fine-grained experts (Dai et al., 2024; Krajewski et al., 2024)), modalities (vision transformer (Zhai et al., 2022)), and other multi-stage training paradigms (as discussed in Section 6) would help assess their generality and implications.

## Software and Data

The source code and data (cross-entropy losses) for analyses of the paper is available at https://github.com/sbintuitions/sparse-upcycling-scaling-laws.

## Impact Statement

This paper works toward the goal of advancing the field of machine learning and language modeling, with an emphasis on scaling. There are many potential societal consequences of our work, none which we feel must be specifically highlighted here.

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

# A. More on Architecture and Experimental Design

## A.1. Megatron-LM configuration

**Infrastructure.** Our experiments are performed on multiple nodes, each consisting of 8 NVIDIA H100 80 GB GPUs, interconnected via InfiniBand HDR. The software we use for training is the Megatron-LM library (Shoeybi et al., 2019).

We use and modify the Megatron-LM (core v0.8.0) library for our experiments[4]. Models are trained with data type bfloat16. Except for the largest MoE we train (8x1B), which has tensor parallelism configured to be 2, all models are trained with data and sequence parallelisms only (Korthikanti et al., 2023). Other optimization libraries used include FlashAttention (Dao et al., 2022) and TransformerEngine[5]. See the example scripts provided on Github[6].

## A.2. Model configuration

Let us elaborate more on our architectural choices. The intermediate hidden dimension size, $d_{\mathrm{MLP}}$, is set to be $4d_{\mathrm{model}}$. We do not implement bias in the linear layers. We also do not use techniques geared for treating training instabilities (which we did not encounter in our study), such as Z-loss or QK normalization. Efficiency-motivated implementations like grouped query attention are not considered as well for simplicity. The number of attention head is chosen to increase with model size following practices in the LLM literature. Other designs of the architecture follow Llama2's closely (Touvron et al., 2023). See Table 3 for the model configurations. They are selected such that the ratio $n_{\mathrm{layer}}/d_{\mathrm{model}}$ lies in the range 32 to 64, as in Kaplan et al. (2020). We use the smaller models for ablation studies.

## A.3. MoE configuration

Let us describe the routing mechanism within the MoE module studied in this work. Denote the number of experts by $n_{\mathrm{expert}}$, the number of activated experts by $n_{\mathrm{TopK}}$, and the output of expert $i$ by $O_{\mathrm{exp},i}$. At each layer, the output tokens $x$ of the attention layer are passed to a router, which consists of a single-layer perceptron with weight $W$, responsible for calculating $(G_1(x), G_2(x), ..., G_{n_{\mathrm{expert}}}(x))$, where

$$G(x) = \mathrm{Softmax}(\mathrm{TopK}(W \cdot x)) \qquad (15)$$

The TopK operation ensures that only $n_{\mathrm{expert}}$ experts are activated for each token, resulting in sparse computation. The output of the MoE module, $O_{\mathrm{MoE}}$, is the weighted expert outputs which can be written as follows:

$$O_{\mathrm{MoE}} = \sum_{i=1}^{n_{\mathrm{expert}}} G_i(x)O_{\mathrm{exp},i}(x)$$

Note that this is also known as token-choice algorithm. Furthermore, we do not use the token-dropping mechanism as in Fedus et al. (2022). We also do not study MoE variants such as shared experts and fine-grained experts (Dai et al., 2024; Krajewski et al., 2024), as upcycling these variants is not straightforward.

Let us move to discussing the load-balancing loss. It has the form (Fedus et al., 2022):

$$\mathcal{L}_{\mathrm{aux}} = \frac{4\eta}{T^2} \cdot \sum_{i=1}^{n_{\mathrm{experts}}} \left( \left( \sum_{j=1}^{T} P_{j,i} \right) \cdot Q_i \right),$$

where $\eta$ is the coefficient for the auxiliary loss, $T$ is the number of tokens, $P_{j,i}$ the router output probability for token $j$ to be assigned to token $i$, and $Q_i$ is the number of token assigned to expert $i$. We ablate the coefficient in Appendix A.7.

## A.4. Ablation of learning rate schedules

Here, we compare the performances of using WSD and the commonly used learning rate (LR) cosine schedules. Dense model and MoE used in our ablation are 0.1B and 8x44m respectively, with training configuration given in Table 6. We can see from Figure 7 that both schedules yield similar performances.

---

[4] https://github.com/NVIDIA/Megatron-LM
[5] https://github.com/NVIDIA/TransformerEngine
[6] dense model example: https://github.com/NVIDIA/Megatron-LM/tree/main/examples/gpt3; MoE example: https://github.com/NVIDIA/Megatron-LM/tree/main/examples/mixtral

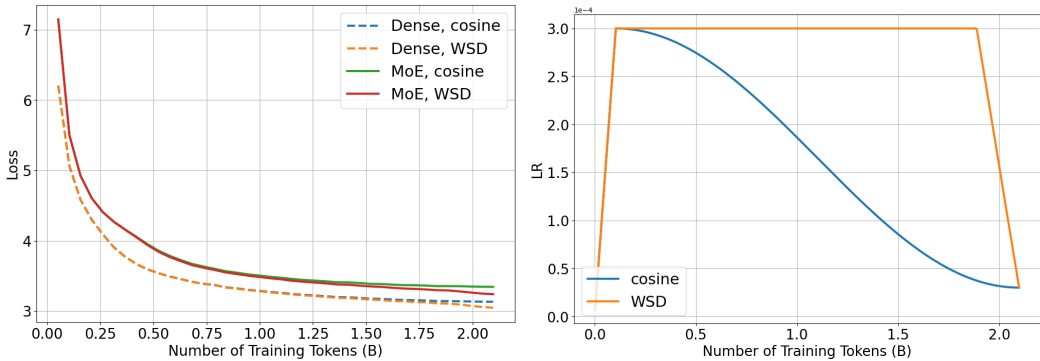

*Figure 7.* **Comparing WSD and cosine schedules of learning rate.** Left: we see that different schedules cause little differences between the losses, for both dense and MoE training. Right: the learning rate schedules in use are shown.

### A.5. Training configuration

The common setup of training is shown in Table 4, and the model-dependent setup (warmup iteration, standard deviation of the normal distribution for initializing weights, maximum iteration run, batch size, tuned LR) is shown in Table 5. As described in the main text, we use the WSD schedule for training. The number of warmup steps of the WSD LR schedule is set to be roughly the same as the total model size (Porian et al., 2024). Linear decay to 10% of the maximum LR value is used in the last stage of the schedule, with the length set to be around 10% of the training length, following Hägele et al. (2024).

Logarithmically-spaced intermediate checkpoints are saved and used to emulate different numbers of training token budget. We also increase both the training length and batch size with model size following common practices without performing precise tuning.

| Model | $n_{\text{layer}}$ | $d_{\text{model}}$ | $n_{\text{head}}$ | $N_{\text{dense}}$ | $N_{\text{MoE}}^{\text{total}}$ (8 experts) |
|---|---|---|---|---|---|
| **15M** | 9 | 320 | 4 | 14,751,680 | 92,189,120 |
| **44M** | 12 | 480 | 8 | 44,248,800 | 276,538,080 |
| **0.1B** | 15 | 640 | 8 | 98,323,840 | 614,496,640 |
| **0.2B** | 21 | 832 | 8 | 232,623,040 | 1,453,845,952 |
| **0.5B** | 26 | 1120 | 16 | 521,889,760 | 3,261,732,320 |
| **1B** | 30 | 1504 | 16 | 1,085,859,424 | 6,786,500,704 |

*Table 3.* Dense models used in our study and their parametric details. Note that $d_{\text{MLP}}$, is set to be $4d_{\text{model}}$. In the last column, we show the total non-embedding model parameters of the corresponding MoE with 8 experts.

| Configuration | Details |
|---|---|
| Context length | 1,024 |
| Embedding tying | False |
| Optimizer | AdamW (Loshchilov et al., 2017) |
| Adam $\beta_1$ | 0.9 |
| Adam $\beta_2$ | 0.95 |
| Adam $\epsilon$ | 1e-8 |
| Weight decay | 0.1 |
| Gradient clipping | 1.0 |

*Table 4.* Training configuration used throughout the paper.

| Model | warmup iter. | init. size | Max iter. | batch size | LR (8x) |
|-------|--------------|------------|-----------|------------|---------|
| **15M** | 200 | 0.035 | 17,600 | 128 | 8e-3 (2e-3) |
| **44M** | 200 | 0.029 | 17,600 | 256 | 4e-3 (2e-3) |
| **0.1B** | 200 | 0.025 | 17,600 | 512 | 4e-3 (2e-3) |
| **0.2B** | 400 | 0.022 | 35,200 | 512 | 2e-3 (2e-3) |
| **0.5B** | 800 | 0.019 | 70,400 | 512 | 4e-4 (4e-4) |
| **1B** | 800 | 0.016 | 70,400 | 1024 | 4e-4 (4e-4) |

*Table 5.* **Model-dependent training configuration**. "init. size" refers to the standard deviation of the normal distribution used for initializing the weights. "Max iter." refers to the maximum iteration run on the model. The MoE counterpart uses the same configuration except for the learning rate (last column).

| Configuration | Details |
|---------------|---------|
| Batch size | 512 |
| train iter. | 4,000 |
| Warmup iter. | 200 |
| Auxiliary loss coeff. | $10^{-3}$ |

*Table 6.* Training configuration for ablation studies.

## A.6. Upcycled training's configuration

We initialize the router weights from a normal distribution with zero mean and variance of $2/5d_{\text{model}}$ (Le Scao et al., 2022; Chowdhery et al., 2023; Zeng et al., 2022) (the same initialization is used for from-scratch trainings).

Regarding the learning rate (LR) of upcycled training, there are several choices: using the LR at the end of dense training with a constant LR schedule, i.e., treating upcycled training as a kind of fine tuning; using the LR of dense/MoE training (Komatsuzaki et al.; He et al., 2024). We consider these three choices without retuning the LR. With the other training settings are set to be the same as the one used in training the dense models, including the use of the WSD LR schedule, we find that using the MoE LR leads to better performance. See Figure 8.

As a side note, we observe that the loss for the constant LR schedule decreases monotonically, while the loss increases initially for other cases; there is a *rewarming* stage when using the WSD LR schedules, which can also be observed in Figure 3.

Let us further comment on alternative upcycling methods (parameter reinitialization) that could potentially further accelerate training. Adding some form of noises to the dense MLP weights, or modifying partially the weights would intuitively help upcycled training generalize faster. Our preliminary experiments (adding Gaussian noise to the expert weights of the upcycled model; partially randomizing the MLP weights; and substituting the MLP weights with their low-rank couterparts randomly to encourage expert divergence) however did not see any advantages of doing so. Note that this observation is also consistent with previous negative reports (Komatsuzaki et al.; Wei et al., 2024; Muennighoff et al., 2024). Henceforth, we simply copy the weights directly from the dense model to perform upcycled training.

## A.7. Ablation of auxiliary coefficients for load-balancing loss

The auxiliary loss affects the load-balancing loss (smaller values meaning more balanced expert usage) as well as the cross-entropy loss. We plot these losses in Figure 9 for training a 8x0.1B MoE.

We see that, setting the coefficient to be too small leads to imbalance in expert usage, while setting the coefficient to be too large interferes with the cross-entropy loss. We set it to around $10^{-3}$ which gives the right balance. We did not make finer tuning of the coefficients and adopted $10^{-3}$ in our experiments.

## A.8. Ablation of dataset repetition for upcycling

In our experiments, we have used the same dataset for training both dense and upcycled models, following practices of (He et al., 2024). As an ablation, we split our dataset to two non-overlapping portions, and perform the two-stage training with

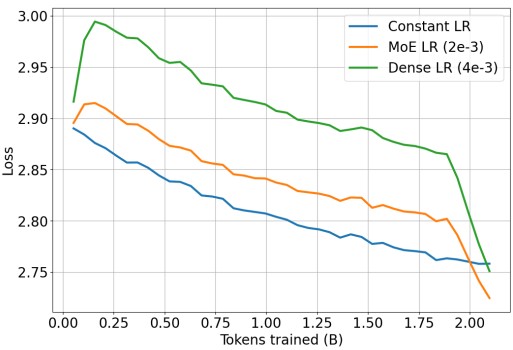

*Figure 8.* **Ablation of LR when upcycling an 8x0.1B MoE.** We compare the performance of upcycled training (from a dense model trained for 2B tokens) using constant LR ($2 \times 10^{-4}$, LR at the end of dense training), LR used for dense training, and LR used for MoE training. We find that the latter works the best. The training setup follows Table 6.

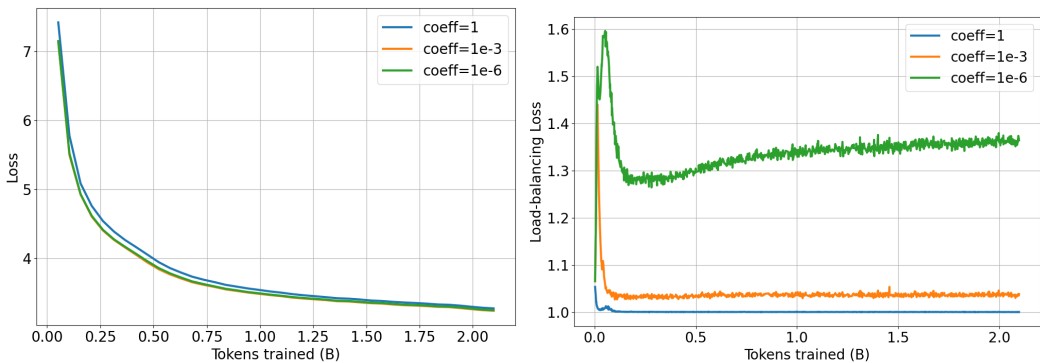

*Figure 9.* **Ablating auxiliary coefficients.** Left: Cross-entropy losses, where it can be seen that auxiliary coefficient of 1 performs worst. Right: Load-balancing losses, where the larger the coefficient is, the smaller the load-balancing loss becomes. Setting the coefficient to be $10^{-3}$ gives the right balance between these two losses.

the distinct datasets. In Figure 10, we find that the difference in performance is very small. This also aligns with previous investigation on data repetition, where it is shown that there is a little difference in performance up to 4 times of repetition (Muennighoff et al., 2023).

### A.9. Evaluation with standard benchmarks and comparison with other existing models

We compare the performance of our trained 1B models against existing models with similar sizes, Pythia (Biderman et al., 2023) and TinyLlama (Zhang et al., 2024), based on standard natural language processing benchmarks, ARC (Clark et al., 2018), lambada (Paperno et al., 2016), logiqa (Liu et al., 2021), piqa (Bisk et al., 2020), sciq (Welbl et al., 2017), and winogrande (Sakaguchi et al., 2021).

Table 7 shows the results. First, we see that the dense model perform similarly to the open models, indicating that our models have been trained correctly. Second, we see that the upcycled model achieves overall the best performance (note that the total tokens used for dense and upcycled training are around 100B, similar to those under comparison), also indicating that upcycling has progressed correctly and improved the scores.

### A.10. Estimated Total GPU hours

Instead of reporting the actual runtimes on our cluster, which varied in our experiments due to many factors affecting the cluster (number of available nodes, congestion, etc.), we give a theoretical estimate of total GPU hours used for obtaining the joint scaling law, which involves running the largest tested model with most training tokens in this paper.

The estimate is as follows. We calculate the FLOPs for training the dense, MoE, upcycled MoE models with maximum

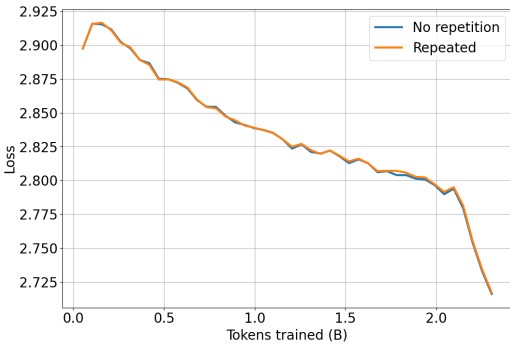

*Figure 10.* **Ablation of data repetition when upcycling an 8x0.1B MoE.** We do not observe notable difference in the loss. The training setup follows Table 6.

| Models | Pythia-1B | TinyLlama-1.1B | 1B | Upcycled 8x1B |
|---|---|---|---|---|
| **Datasets** | Pile | Slimpajama & Starcoder | Slimpajama | Slimpajama |
| **Tokens** | 100B | 103B | 74B | 37B |
| ARC-c | 25.59 | 24.32 | 27.65 | **30.12** |
| ARC-e | 47.26 | 44.91 | 52.10 | **56.14** |
| lambada | **53.52** | - | 45.08 | 49.72 |
| logiqa | **29.49** | - | 26.11 | 27.65 |
| piqa | **69.31** | 67.30 | 65.89 | 67.19 |
| sciq | 77.3 | - | 78.10 | **82.00** |
| winogrande | 51.22 | 53.28 | 54.93 | **57.77** |
| **Avg.** | 50.53 | - | 49.98 | **52.94** |

*Table 7.* **Benchmarks' performance comparison across models.** Reported scores are accuracies (normalized by byte length whenever applicable). The first two columns are scores of existing models. The last two columns are evaluation results of models trained in this work. The upcycled 8x1B model is upcycled from the 1B model. Our models are evaluated with the LM Evaluation Harness v0.4.0 library (Gao et al., 2024).

iterations using the $6ND$ approximation, ignoring the additional FLOPs required to continued pretrain models with shorter iterations (as we can reuse the intermediate checkpoints). We further assume that the per-second TFLOPs of the GPU is 400, and is the same for both dense and MoE models [7]. We obtain,

**Dense model:** $6.14 \times 10^{17}$ FLOPs

**MoE:** $1.08 \times 10^{18}$ FLOPs

**Upcycled MoE:** $5.38 \times 10^{18}$ FLOPs

The total GPU hours are henceforth approximately 4,900. We note that the exact total GPU hours used are larger as we have run various additional studies as detailed in the paper. We further note that using the cosine learning rate schedule would cost about twice more GPU hours when varying the number of training token budget.

## B. Generalization across Other Datasets

We show that the scaling behavior studied in the main text generalizes across datasets in use. Aside from Slimpajama (Computer, 2023; Soboleva et al., 2023; Shen et al., 2023) used in the main text, we test with two additional datasets: Japanese portion of the CommonCrawl corpus, and the Stack code dataset (Kocetkov et al.). We use a tokenizer of enlarged

---
[7]https://github.com/NVIDIA/Megatron-LM/blob/main/megatron/core/transformer/moe/README.md.
Note that MoE requires more GPU memory to store its total model parameters, inducing overhead that may slow down training.

vocabulary size of 102,400 when training with the former dataset [8]. We upcycle a 0.1B model with the training configuration set to the same as those used for the experiments presented in the main text. The results are presented in Figure 11, where the scaling behaves similarly to those presented in Figure 4, suggesting that our scaling law for dataset is independent of dataset in use. We however do not further study scaling law for model configuration and the joint scaling law due to computational constraints.

We further observe:

1. The Japanese dataset has higher validation loss (harder task), while the code dataset results in lower loss, making it a relatively easier task in terms of cross-entropy loss.

2. Interestingly, the Japanese dataset is harder to saturate with increasing $D_1$, meaning upcycled training remains effective. In contrast, the code dataset saturates more quickly, making upcycling less beneficial.

The more fine-grained question, e.g., how mixtures of pretraining data impact respective downstream performances, is beyond the scope of our current work, and we leave it for future study.

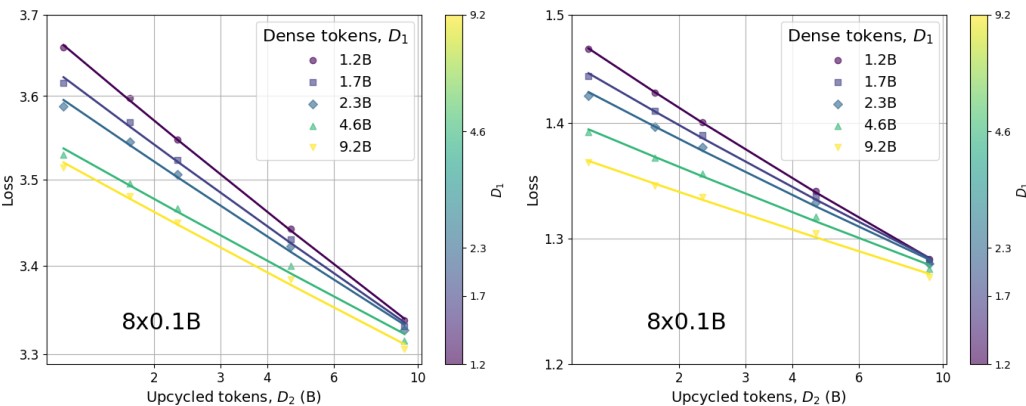

*Figure 11.* **Scaling behavior generalizes across datasets used.** Left: Japanese dataset. Right: Code dataset.

## C. More Results on the Scaling Law for Dataset Sizes

### C.1. Fitting across architectures

Figure 8 shows the fitting of the scaling law for dataset sizes across architectures (fixing dense model size to 0.1B).

| $(n_{\text{expert}}, n_{\text{TopK}})$ | (4,1) | (4,2) | (8,1) | (8,2) |
|---|---|---|---|---|
| $\alpha_1$ | 0.28 | 0.19 | 0.29 | 0.18 |
| $\alpha_2$ | 0.29 | 0.20 | 0.30 | 0.20 |
| $\alpha_3$ | 0.01 | 0.008 | 0.01 | 0.008 |

*Table 8.* **Fitted exponents of the scaling law for dataset sizes across architectures.**

### C.2. More validation of Equation 1

We first make an observation from Figure 4 that the slope of the fitted lines, that is, the scaling exponent as in Equation 7 is decreasing with $D_1$. Let us consider the scaling factors $A$ fitted in Figure 4. The scatter plot of $A$'s in Figure 12 leads us to deduce that the following relation:

$$A \propto D_1^{-\eta} \tag{16}$$

---

[8] https://huggingface.co/sbintuitions/sarashina2-70b

Indeed, we find that the above equation fits well in the Figure. Substituting it to Equation 7, we obtain at the RHS a term of the form $D_1^{-\eta} D_2^{-\alpha}$, indicating that Equation 1 arises not only from first principles as in the main text, but from empirical observations as well.

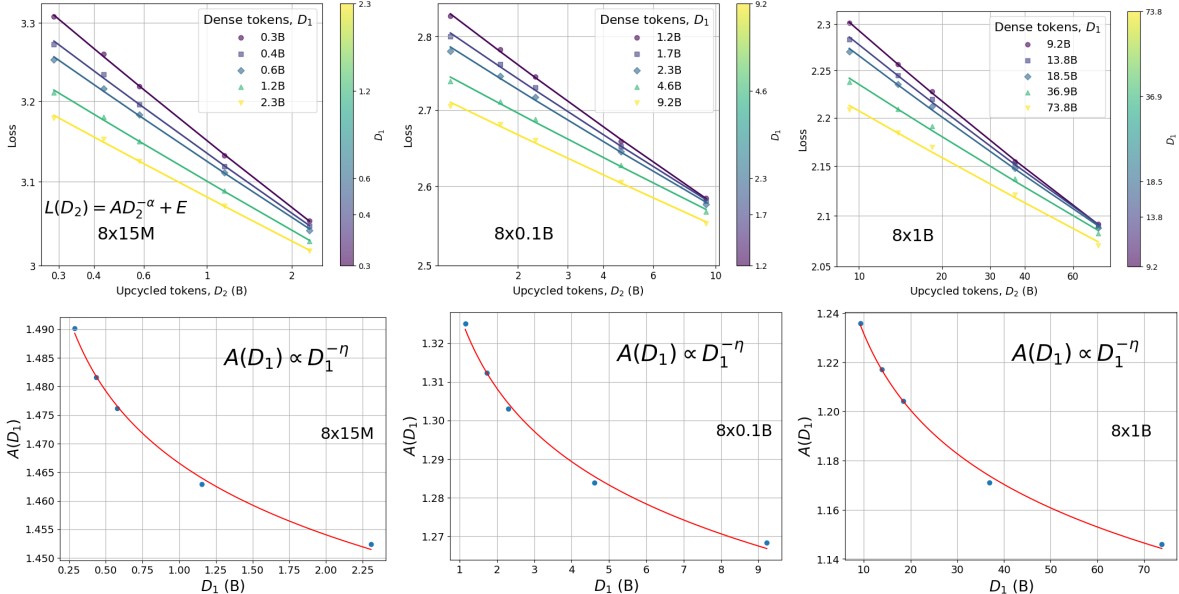

*Figure 12.* **Top: Fitting plots same as those in Figure 4. Bottom: Fitting scaling factor has power-law behavior.** The fitted scaling factors are shown to fit well with Equation 16.

### C.3. Bilinear form

We would also like to note that by taking the logarithm of the first term of Equation 8, we have $\alpha_1 \log D_1 + \alpha_2 \log D_2 + \alpha_3 \log D_1 \log D_2$, which is *bilinear* in terms of $\log D_1$ and $\log D_2$ (that is, linear with each variable separately), indicating that the multiplicative term with interaction is a natural generalization of the single-variable power law.

## D. More Results on the Scaling Law for Model Configuration

### D.1. Fitting the Scaling Law for Model Configuration

Recall that we wish to understand the cross-entropy loss as a function of sparsity, defined as $P = N_{\text{total}}/N_2$ and the number of active parameters, $N_2$. Starting from the power-law ansatz, we require that the loss satisfies

$$L(P, N_2) = L_P(N_2) = A N_2^{-\beta_1} + E$$

and

$$L(P, N_2) = L_{N_2}(P) = A P^{-\beta_2} + E$$

This is reasonable as $P$, when fixing $N_2$, is the total number of model parameters, which we expect to satisfy the power-law ansatz. Analogously, $N_2$ corresponds to the number of dense model parameters, which should satisfy the power-law ansatz as well.

We then consider, as before, both additive and multiplicative functional forms, with and without interaction, satisfying the above requirements, and evaluate them using leave-one-out RMS error:

**Multiplicative:**

$$L(P, N_2) = A P^{-\beta_1} (1 + N_2)^{-\beta_2 + \beta_3 \log P} + E$$
$$\approx A P^{-\beta_1} N_2^{-\beta_2 + \beta_3 \log P} + E \tag{17}$$

**Additive:**

$$L(P, N_2) = AP^{-\beta_1} + F(1 + N_2)^{-\beta_2 + \beta_3 \log P} + E$$
$$\approx AP^{-\beta_1} + FN_2^{-\beta_2 + \beta_3 \log P} + E \tag{18}$$

We obtain RMS errors of 0.0414, 0.0351, 0.0341 and 0.0322 for multiplicative (with interaction), multiplicative (without interaction), additive (with interaction), and additive (without interaction) cases, respectively. That is, the additional form without interaction functional form provides the best fit to the data.

### D.2. Alternative Visualization of the Scaling Law for Model Configuration

We show the 2D slices (with fixed sparsity) of the model configuration scatter plot as in the left panel of Figure 1, in Figure 13.

## E. More Results on the Joint Scaling Law

### E.1. Validating Requirement 3

We show in Figure 14 that the upcycled model empirically satisfies power-law scaling with respect to $N_1$, i.e., Requirement 3, repeated below for convenience.

**Requirement 3.** $L_{D_1, D_2}(N_1) = B_{D_1, D_2} N_1^{-\beta} + E_{D_1, D_2}$.

### E.2. Another functional form satisfying Requirements 3 and 4

For convenience purposes, let us repeat Requirement 4:

**Requirement 4.** $\lim_{D_2 \to 0} L(D_1, D_2, N_1) = AD_1^{-\alpha} + BN_1^{-\beta} + E$, i.e., approaching Chinchilla scaling law.

There is another functional form satisfying Requirement 3 and 4:

$$L(D_1, D_2, N_2) = \frac{A}{D_1^{\alpha_1}} + \frac{B}{D_2^{\alpha_2} N_1^{\beta_2 - \alpha_3 \log D_2}} + E \tag{19}$$

Fitting the above equation however yields negative exponents, which are, empirically, $\alpha_1 = 0.10, \alpha_2 = -0.07, \beta = -0.08, \alpha_3 = -0.008$. We reject the hypothesis of this functional form as the loss is predicted to increase with dataset/model sizes, violating the power-law ansatz as well as empirical observation.

### E.3. Why fitting scaling laws separately?

We provide reasoning on why we fit three scaling laws separately, instead of fitting an unified scaling law which holds for all three dense, MoE from-scratch, and upcycled training scenarios.

The scaling law for MoE trained from scratch is expected to have parameters different from those for upcycled training (that is, they are not equal to each other at $D_1 \to 0$), because upcycled MoEs are initialized with identical MLP/expert weights from the dense models, while MoEs trained from-scratch have different initial (random) MLP/expert weights. Thus, we expect them to scale differently.

Similarly, we expect the scaling law for dense model has parameters different from those for upcycled training at the limit $D_2 \to 0$. As can be seen from Figure 3, the upcycled training undergoes a *rewarming* phase in the beginning, where the loss increases initially before decreasing. This causes a deviation from the dense model scaling law at $D_2 \to 0$, although we still expect it to correlate with the dense model scaling law, i.e., Requirement 2 should hold based on function preservation and (empirical) observations that rewarming does not completely de-correlate the loss behaviors.

Finally, our preliminary investigations also show that unified scaling law does not fit well. We consequently consider them separately. Moreover, we note that there can be other possibilities of functional forms that we do not explore further.

# F. More Implication of the Joint Scaling Law

### F.1. From-scratch training vs Upcycling

We have shown in the main text that upcycling is only effective when the sunk cost or the model size is small. We further visualize this by plotting the losses for model sizes 1B, 7B and 70B, fixing $D_1$ to various values, with respect to $D_{\text{MoE}}$ and $D_2$ for from-scratch and upcycled training respectively in Figure 15.

### F.2. More on compute optimality of upcycling

As mentioned in the main text, while training the Mixtral-like MoE from scratch is shown to be more efficient in the long run, it requires more compute (1.75 times larger than dense training). There is a compute-performance trade-off between these stages of training.

We show in Figure 16 that the optimization results in Section 5.2 that training from scratch can always be considered to be more performant.

### F.3. Derivation of compute-optimal formula

As in the main text, we want to scale $N_2, D_2$ optimally, $N_2^{\text{opt}}, D_2^{\text{opt}}$, given a FLOPs budget and fixing $D_1$, while minimizing the loss $L$, which we write as $L_{D_1}(D_2, N_1)$. This is equivalent to solving the following:

$$\frac{\partial}{\partial D_2} L_{D_1}(D_2, C_2/10.5D_2) \bigg|_{D_2=D_2^{\text{opt}}} = 0,$$

$$\frac{\partial}{\partial N_1} L_{D_1}(C_2/10.5N_1, N_1) \bigg|_{N_1=N_1^{\text{opt}}} = 0$$

where we have used $N_2 = 1.75N_1$ and $C_2 = 6N_2D_2$. Solving the above equations leads to

$$D_2^{\text{opt}} = G \left( \frac{C_2}{10.5} \right)^a,$$

$$N_1^{\text{opt}} = G^{-1} \left( \frac{C_2}{10.5} \right)^b$$

where

$$G := \left( \frac{A_{\text{eff}} \alpha_{\text{eff}}}{B\beta} \right)^{1/(\alpha_{\text{eff}}+\beta)}$$

$$a := \frac{\beta}{\alpha_{\text{eff}} + \beta}$$

$$b := \frac{\alpha_{\text{eff}}}{\alpha_{\text{eff}} + \beta}$$

$$A_{\text{eff}} := AD_1^{-\alpha_1}$$

$$\alpha_{\text{eff}} := \alpha_2 - \alpha_3 \log D_1$$

We can henceforth relate $D_2^{\text{opt}}$ and $N_1^{\text{opt}}$ via

$$D_2^{\text{opt}} = G \left( G N_1^{\text{opt}} \right)^{a/b} \propto \left( N_1^{\text{opt}} \right)^{\beta/\alpha_2 - \alpha_3 \log D_1}$$

and

$$N_1^{\text{opt}} = G^{-1} \left( G^{-1} D_2^{\text{opt}} \right)^{b/a} \propto \left( D_2^{\text{opt}} \right)^{(\alpha_2 - \alpha_3 \log D_1)/\beta}$$

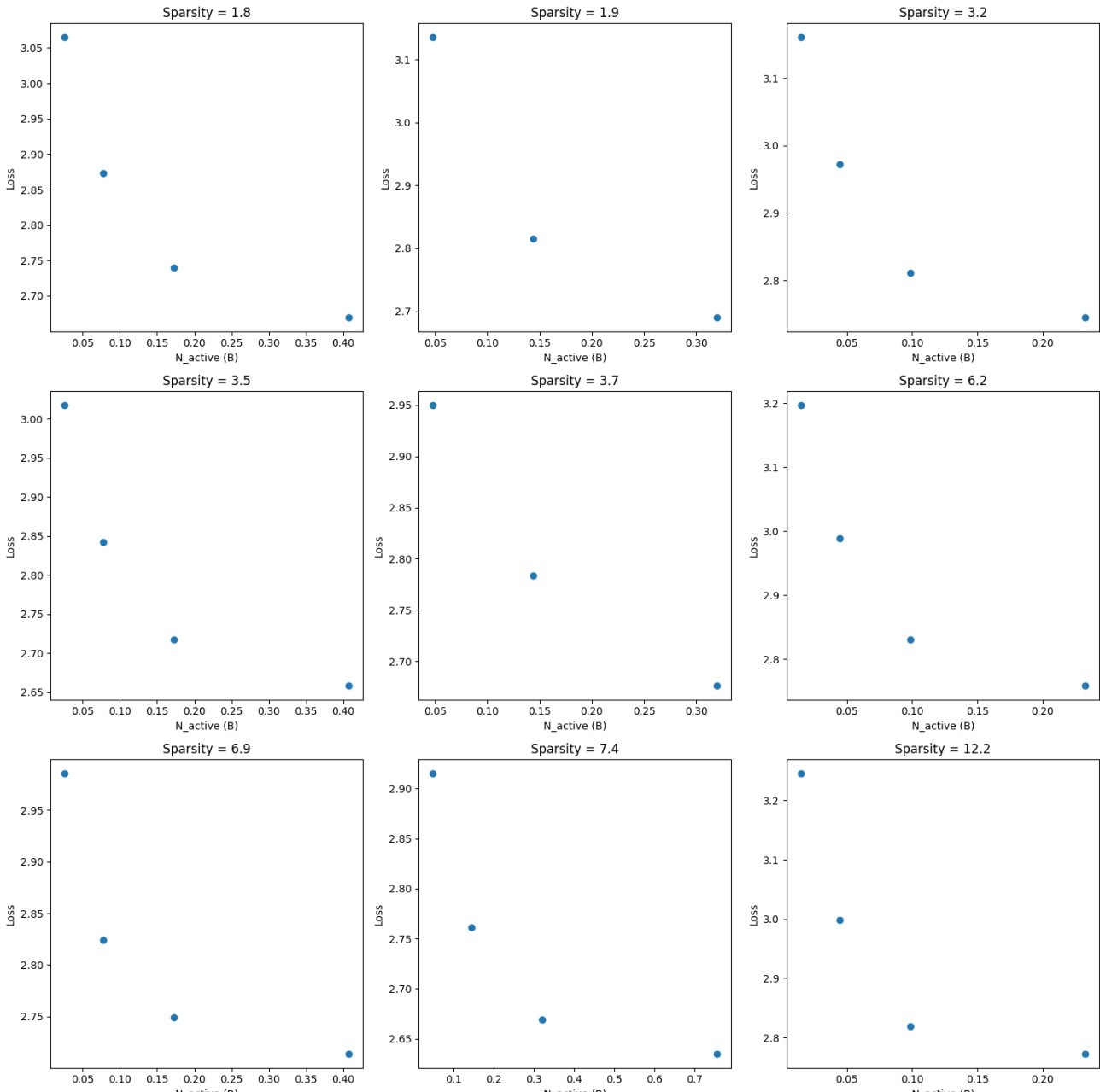

*Figure 13.* **Scaling behavior with respect to model configuration.** Shown are plots of the cross-entropy loss with respect to active parameter $N_2$ fixing sparsity $P$.

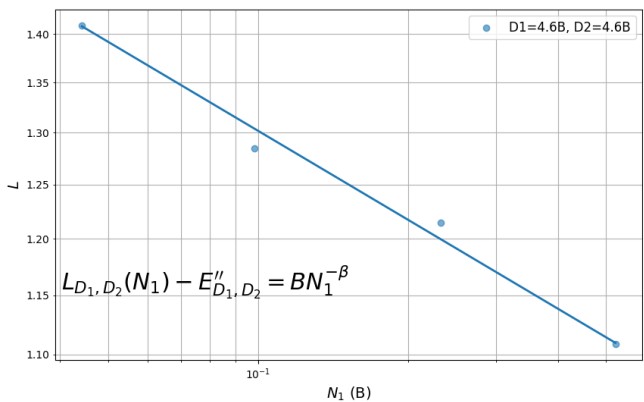

*Figure 14.* **Upcycled model has power-law behavior with respect to $N_1$.**

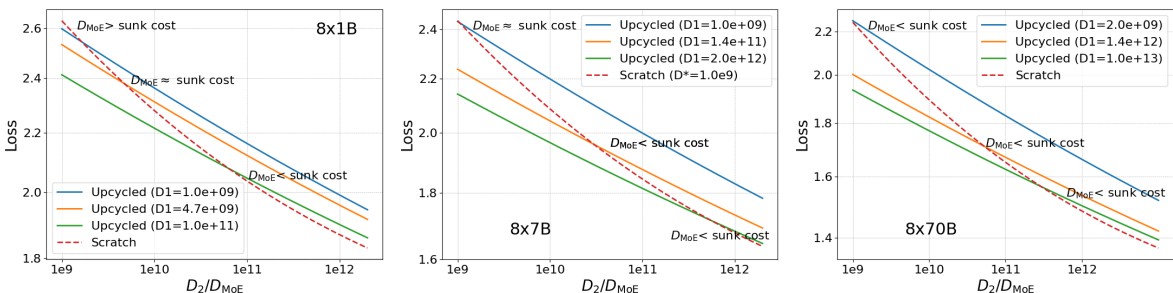

*Figure 15.* **Token budget of from-scratch MoE training and when it catches up with upcycled MoE's performance.** We compare loss-versus-token plots of from-scratch and upcycled MoE trainings at various dense training budgets (sunk costs) and model sizes. Upcycling is considered to be efficient only when $D_{\text{MoE}} >$ sunk cost. We observe that the efficiency of upcycling diminishes with sunk cost and model size.

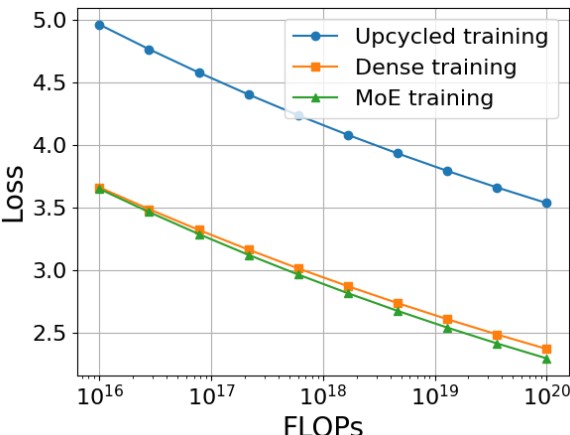

*Figure 16.* **Compute-optimal training.** Upcycled training performs worse even when $D_1, D_2, N_1$ are allocated optimally, compared to compute-optimal training of dense or MoE models.

