# OpenReview forum: "Scaling Laws for Upcycling Mixture-of-Experts Language Models"
_ICML.cc/2025/Conference — ICML 2025 poster_

### Official Review · Reviewer_byMA · 2025-03-11

**Overall Recommendation:** 3

**Summary:**

This paper studies the computationally efficient training of large language models (LLMs) through upcycling where smaller pretrained dense models are utilized as initial checkpoints to train larger Mixture-of-Experts (MoE) models. Given that training large-scale language models from scratch demands considerable computational resources, the authors explore empirical scaling laws related to upcycling dense checkpoints into MoE models which remains relatively unexplored. Through experiments involving models of varying sizes (up to 7B parameters) and datasets containing hundreds of billions of tokens, the authors establish empirical scaling laws that characterize how model performance (measured by cross-entropy loss) depends on factors such as dataset size, number of tokens used in dense pretraining (D1), tokens used in upcycled MoE training (D2), model size, and model sparsity (ratio of total parameters to actively-used parameters). More specifically, a novel multiplicative scaling law is proposed, which describes the MoE model’s cross-entropy loss as a function of both the dense pretrained tokens (D1) and additional MoE training tokens (D2). This scaling law incorporates a previously unidentified interaction term between the dense and MoE datasets, highlighting diminishing efficiency of upcycling as the initial dense training budget (the sunk cost) increases. The authors also identify that the performance of upcycled MoE models consistently improves with increased sparsity and number of active parameters. The paper offers explicit guidance, suggesting scenarios under which upcycling is beneficial or not. For instance, they derive a threshold token count (D*) as a function of dense model size, providing a practical rule to decide whether upcycling is computationally advantageous.

**Claims And Evidence:**

Yes, overall, the claims made in the submission are supported by experimental evidence. Empirical scaling laws for the interaction between dense pretraining tokens (D1_1​), upcycled MoE tokens (D_2​), and model configurations are convincingly supported by experimental results.

**Essential References Not Discussed:**

NA

**Experimental Designs Or Analyses:**

The experimental designs and analyses are sound and valid.

**Methods And Evaluation Criteria:**

Yes, the methods and evaluation criteria make sense.

**Other Comments Or Suggestions:**

Discussing the similarities and differences between the scaling laws in these different settings can make the contribution more impactful and of interest to a wider audience.

Figure 1 (left): it is not easy to see the cross-entropy values, it might be clearer to include a color scale for these values.
While scaling laws for upcycling MOE is novelty, there are previous work on related application such as transfer learning.

**Other Strengths And Weaknesses:**

Strengths:
+ The paper addresses a relatively under-explored topic: the scaling laws specific to "upcycling" dense pretrained models into Mixture-of-Experts (MoE) architectures.
+ The writing is clear, well-organized and easy to follow.
+ The provided scaling laws and guidelines offer practical insights for training large models efficiently, which is highly relevant to the community given the computational expense of training language models.

Weaknesses:
- Limited Generalizability to Larger Models: The experiments are limited to models of size up to 7B parameters, it remains unclear if the results extend to significantly larger MoE models (e.g., 70B+ parameters).
- Lack of Theoretical Explanation
- Scope seems limited to only upcycled MoE models.

**Questions For Authors:**

How do the scaling laws here compare with other scaling laws in related setting such as transfer learning?

**Relation To Broader Scientific Literature:**

The scaling laws derived from this work can impact mixture of experts (MOE) models in diverse domains. However, it is likely limited to only upcycled MOE models.

**Theoretical Claims:**

NA, the paper does not provide explicit theoretical proofs for its claims.

---

> ### Author Rebuttal · Authors · 2025-04-01
>
> We sincerely thank the reviewer for the thoughtful and positive evaluation of our work. We especially appreciate the **recognition of its novelty and relevance, as well as the clarity of our presentation**.
> We also appreciate the constructive suggestions.
> Let us clarify some of the questions and comments raised in the following.
> # Question on Relevance to transfer learning and other scaling laws
> We have briefly discussed connections to transfer learning (and cited relevant two-stage training methods like model growth and pretraining/SFT) in **Section 7**, but we agree that this deserves a more in-depth discussion, which we describe below.
> - Prior work (e.g., Mikami et al 2022) on transfer learning has proposed scaling laws using multiplicative or additive forms involving $D_1$ (pretraining data) and $D_2$ (fine-tuning data).
> - However, as far as we know, there has been no attempt to incorporate an interaction term such as ours, namely $D_1\log D_2$. We believe this interaction is **novel and meaningful**: it is both empirically validated and theoretically motivated by our derivation in **Section 4.1** (see also **Appendix C.3**).
> - While our focus is on upcycling MoE models, we expect the interaction term to also be applicable in transfer learning settings. For example, transfer learning literature has observed *ossification* effects (Hernandez et al 2021), where models pretrained on large datasets can become resistant to adaptation during fine-tuning.
> - The diminishing returns we observe from increasing $D_1$ align with this notion. Our interaction term $D_1 \log D_2$ captures precisely such diminishing gains and may offer a useful framework for modeling ossification quantitatively in transfer learning scaling laws (and other two-stage training methods).
>
> We will revise the discussion to emphasize this connection.
>
> # Comment on Limited Generalizability to Larger Models
> - We acknowledge the concern about generalizability to larger models; however, we note that this limitation is common across empirical studies of scaling laws, including highly influential ones.
> - Training and evaluating models at the 70B+ scale requires orders of magnitude more computational resources which is currently infeasible in an academic setting. We estimate this to require 5,000 times more FLOPs (70x larger model and 70x more tokens) than what we have run (see Appendix A.10), amounting to an additional cost of 25 million USD (even assuming 1 USD per GPU hour) .
> - Despite this, we provide empirical evidence supporting the robustness of our findings: in **Figure 6**, we show that scaling behavior observed in sub-0.5B parameter models can **reliably predict** the performance of larger models (up to 1B parameters).
> - This suggests that the scaling laws identified are stable across model sizes, and we expect the trends to extend to even larger models, consistent with patterns observed in prior work (e.g., Chinchilla paper).
>
> # Comment on theoretical explanation
> - While we indeed have not included formal theoretical proofs, we emphasize that the proposed scaling law is not purely empirical. Its functional form is guided by well-motivated principles, as detailed in **Section 4.1**, and was **noted by two other Reviewers to be a reasonable theoretical contribution to empirical scaling law studies**.
> - Our work also follows the tradition of impactful scaling law work (e.g., OpenAI, Chinchilla paper) that focused on empirical observations and practical utility, without extensive theoretical justification.
> - That being said, we agree that theoretical understanding is important, and although it is not the primary focus of this paper, we provide related references in **Section 7**.
>
> # Comment on Limitation to upcycled models
> We respectfully disagree with the characterization that our work is limited in scope due to its focus on upcycled MoE models.
> - In practice, both MoE architectures and upcycling strategies have become central components of modern LLM development. Recent state-of-the-art models, including DeepSeek, Qwen, Skywork-MoE, and Mixtral, adopt MoE architectures while also leveraging dense-to-sparse upcycling. This growing trend highlights that upcycling is not just a research curiosity but a practical and widely adopted technique in the current LLM landscape. Our findings are therefore both timely and relevant, offering insight into how/when upcycling can be efficient.
> - Although our experiments specifically focus on upcycling into MoE models, the core insights, such as the interaction between dense and upcycled training budgets, are useful for formulating two-stage training regimes more broadly, including transfer learning as mentioned above, and potentially model growth and pretraining-SFT framework as cited in the main text.
>
> We hope these address your concern. Please let us know if further clarification would be helpful.

---

### Official Review · Reviewer_j1Qr · 2025-03-11

**Overall Recommendation:** 3

**Summary:**

This work investigates the scaling behavior of upcycling dense LLMs into mixture-of-experts architectures. Through extensive experiments, the authors design and fit scaling laws that describe how language modeling performance depends on dataset size and MoE configuration, including sparsity and number of experts. They find that while upcycling can reduce initial losses and accelerate convergence, its efficiency diminishes as the size of the dense pretrained model and upcycled dataset size increase. Beyond a certain computational budget, from-scratch MoE training becomes more effective than upcycling. The study provides a quantitative framework for evaluating when and how to adopt upcycling and offers guidance on scaling dataset size and model configuration for efficient pretraining.

**Claims And Evidence:**

The claims in this work, specifically the empirically determined scaling laws, are backed up by good experimental evidence.

**Essential References Not Discussed:**

None that I am familiar with.

**Experimental Designs Or Analyses:**

I think the experimental design (train dense models at various sizes with varying token budgets, then upcycle to MoEs with varying token budget and various number of experts and sparsity) is very appropriate.

**Methods And Evaluation Criteria:**

The use of a Llama-like architecture makes sense for evaluating scaling laws. As does the use of SlimPajama, a dataset composed of a number of commonly used subdomains.

**Other Comments Or Suggestions:**

- I think Figure 3 can be made more clear. At the moment, it appears that only 2 of the lines (D2 = 9.23) have the full training curve. For example, presumably, the red line (D1=9.23, D2=4.61) has been trained for 4.61 billion tokens after upcycling, so why does the curve only start around 4.1 billion?
- It would be very interesting to see how the scaling laws of different text domains are similar/different. Since the SlimPajama dataset is composed of a number of domains, this should theoretically be possible with your dataset. In particular, I think it would be very interesting to see how the mixture of pre-training data impacts the downstream performance on each individual domain

**Other Strengths And Weaknesses:**

Strengths:
- This work provides extensive experimentation and demonstrates the quality of their fitted scaling laws for multiple settings
- The exploration of multiple functional forms makes the results more sound

Weaknesses:
- While this work discusses a threshold at which it is better to train an MoE from scratch, there is no clear guidance on how to predict or calculate this threshold in practice for new settings. This area could benefit from more exploration and a deeper analysis

**Questions For Authors:**

- This finding “Increasing D1 (sunk cost) reduces the initial losses of the upcycled MoE but results in slower training progress with D2 (upcycled MoE tokens).” sounds like the initial parameterization of the upcycled MoE is already in a local minima. Are there possible techniques that you may consider to account for this finding? For example, would adjusting the initialization of each expert in the upcycled model allow the experts to diverge quicker and improve the scaling law?

**Relation To Broader Scientific Literature:**

The experiments in this work are interesting, but also very narrow. The authors implement only the most basic form of upcycling. If the authors explored other methods of initializing the upcycled MoE, this work could be of interest to a wider community

**Theoretical Claims:**

I’m not an expert in scaling laws, but the functional forms of the scaling laws they fit for both dense, and MoE models makes sense at a high level.

---

> ### Author Rebuttal · Authors · 2025-04-01
>
> We sincerely thank the reviewer for the thoughtful and positive evaluation of our work. We are especially grateful for the **recognition of our experimental design, the quality of the fitted scaling laws, and the clarity of our empirical methodology**.
> We also appreciate the constructive suggestions.
> Let us clarify some of the questions and comments raised in the following.
>
> # Question on initial parameterization
> We appreciate the reviewer’s insightful observation on whether modifying initialization can improve performance, and indeed we have studied this and mentioned it in **Appendix A.6**. Let us elaborate on it further:
> - To investigate whether this effect could be mitigated, we experimented with several initialization modifications for the MoE experts. These included (i) adding Gaussian noise to the expert weights of the upcycled model, (ii) partially randomizing the MLP weights, and (iii) applying low-rank approximations to encourage expert divergence. However, across all these approaches, we observed no significant improvement in training dynamics or final performance.
> - These results are consistent with findings from prior work, including Komatsuzaki et al (2022), He et al (2022), and Muennighoff et al (2023), who similarly explored various re-initialization or weight-perturbation strategies in MoE settings, with limited success. Overall, our experiments support the conclusion that the vanilla upcycling strategy (without additional modification) is the most effective approach currently known.
>
> # Comment on Figure 3
> The reason some training curves begin from intermediate number of tokens (e.g., around 4.1 B tokens) stems from our use of the Warmup-Stable-Decay (WSD) learning rate schedule combined with a checkpoint reuse strategy for efficiency (explained in **Section 3.1**).
> - Specifically, for runs with smaller $D_2$ budgets (e.g., $D_2$ = 4.61B), we initialize from an intermediate checkpoint of a longer run ($D_2$ = 9.23B) and continue pretraining from that point to reach the desired total token count, without needing to retrain from scratch for every token budget.
> - Therefore, the curve that appears to start around 4.1B tokens means that it has already undergone 4.1B tokens of training along the “main branch”, and the red line shown corresponds to additional continued pretraining to reach a total of 4.61B tokens.
>
> # Comment on different text domains
> Thank you for the insightful suggestion. We agree that understanding how scaling laws vary across text domains is an important direction. In fact, we already have some results on this in **Appendix B**, where we replicate our experiments on two additional datasets, Japanese and code, to test the generality of our findings. Let us elaborate:
> - We observe that the scaling relations introduced in the main text hold consistently, regardless of dataset.
> - The Japanese dataset has higher validation loss (harder task), while the code dataset results in lower loss, making it a relatively easier task in terms of cross-entropy loss.
> - Interestingly, the Japanese dataset is harder to saturate with increasing $D_1$, meaning upcycled training remains effective. In contrast, the code dataset saturates more quickly, making upcycling less beneficial.
>
> This is a promising evidence that the core scaling behavior we identify is robust across diverse domains.
> However, the more fine-grained question of how mixtures of pretraining data impact respective downstream performances is an orthogonal direction.
>  While interesting, it is beyond the scope of our current work.
> For readers interested in downstream impacts, we do include results in **Appendix A.9 and Table 7**, where we evaluate downstream performance on various tasks for models trained with SlimPajama.
>
> # On analyzing the threshold
> - In **Section 5.1**, we define the threshold $D^*$ as the token count at which training a model from scratch matches the performance of an upcycled MoE with the same total token budget using derived scaling laws with fixed model configuration. We solve this equation numerically and also provide an analytic approximation (Equation 2) to aid practical use.
> - We also note in the text that covering all possible configurations and settings would require **exponentially more compute**, which is infeasible in academic environments. As such, we focus on a widely used MoE configuration (Mixtral) to make the analysis tractable.
> - Finally, we offer guidance on how to apply this threshold in practice, highlighting the need to balance model size, compute, and token budgets in the beginning of **Section 5 and Section 7**. Once the configuration is set, one can run the scaling law experiments and use the procedure mentioned above to get the threshold.
> We will revise the text to make these points clearer.
>
> We hope these addresses your concern. Please let us know if further clarification would be helpful.

---

### Official Review · Reviewer_wa23 · 2025-03-13

**Overall Recommendation:** 3

**Summary:**

This paper investigates the scaling laws for upcycling pretrained dense language models (LLMs) into sparse Mixture-of-Experts (MoE) architectures. By conducting extensive experiments with models up to 1B dense and 7B MoE, the authors identified scaling laws which describe the relationship between the cross-entropy loss with dataset size and model configuration. This paper also indicate that upcycling is more efficient than training from scratch under certain conditions.

**Claims And Evidence:**

1.	In section 4.1, the derivation of scaling law for dataset sizes is convincing. The authors firstly set some priori requirements for the functional form of the scaling law (Requirement 1 and 2, which are reasonable), and then empirically fit the functional form with experiment results.
2.	In section 4.2, the hypothesized functional form (Equation 11) requires stronger justification. It is not clear why the relationship between the loss and model configuration should follow the form of Equation 11. The paper does not provide theoretical motivation or ablation studies comparing alternative formulations.
3.	In section 5.2, the authors claimed that larger pretrained models require disproportionately more tokens for efficient upcycling, and that upcycling is inefficient relative to from-scratch trainings when considering compute optimality. These claims are supported by the derived scaling law. However, concrete experiments would significantly strengthen these claims.

**Essential References Not Discussed:**

no.

**Experimental Designs Or Analyses:**

The experimental designs and analysies are largely sound for the studied scope (models up to 1B dense / 7B MoE).

**Methods And Evaluation Criteria:**

The proposed methods and evaluation criteria are appropriate with the problem of understanding upcycling efficiency and scaling laws for MoE models. For example, using WSD LR schedule can reduce computation overhead, and its reliability is verified in the appendix.

**Other Comments Or Suggestions:**

1.	Equation 12: correct left-hand side to L(D1, D2, N1).
2.	Figure 1: replace the 3D plot with 2D slices (fixed sparsity/parameters) for better readability.

**Other Strengths And Weaknesses:**

Strengths:
1.	First systematic study of upcycling scaling law for MoE models.
2.	This work provides practical insights for MoE model training.
3.	Code is available.
Weaknesses:
1.	Some claims may need empirical support. See item 3 in Claims And Evidence.

**Questions For Authors:**

See weakness.

**Relation To Broader Scientific Literature:**

This paper appropriately cites relative works on MoE architecture, scaling law, and upcycling.

**Theoretical Claims:**

This work adopts an empirical approach, which aligns with the practical approaches of empirical scaling law research. Some  theoretical analysis seem reasonbale.

---

> ### Author Rebuttal · Authors · 2025-04-01
>
> We sincerely thank the reviewer for the thoughtful and positive evaluation of our work.
> We especially appreciate the **recognition of our empirical approach, the soundness of our experimental design, and the validation of our scaling law derivation in Section 4.1**.
> We also thank the reviewer for acknowledging the practical value of techniques such as the WSD learning rate schedule.
> The comments regarding the justification of Equation 11 and the desire for more concrete experiments around compute optimality are very helpful, and we address them in detail below.
> # Comment on Equation 11
> We acknowledge that a derivation and ablation study of Equation 11 were lacking, an oversight on our part. We appreciate the opportunity to address it.
> - During the rebuttal period, we conducted a principled analysis to derive and validate the appropriate functional form similar to what is done in section 4.1.
> - Recall that we wish to understand the cross-entropy loss as a function of sparsity, defined as $P= N_{\rm total}/N_2$ and the number of active parameters, $N_2$.
> - Starting from the power-law ansatz, we require that the loss satisfies $L(P,N_2)= L_{P}(N_2) = A N_2^{-\beta_1} + E$ and $L(P,N_2)= L_{N_2}(P) = A P^{-\beta_2} + E$. This is reasonable as $P$, when fixing $N_2$, is the total number of model parameters, which we expect to satisfy the power-law ansatz. Analogously, $N_2$ corresponds to the number of dense model parameters, which should satisfy the power-law ansatz as well.
> - We then consider, as before, both additive and multiplicative functional forms, with and without interaction, satisfying the above requirements, and evaluate them using leave-one-out RMS error. We obtain
>
> | multiplicative (with interaction)           | multiplicative (without interaction)         | additive (with interaction)         | additive (without interaction)       |
> |------------------|----------------|------------------|----------------|
> | 0.0414 | 0.0351 | 0.0341 | 0.0322 |
>
> i.e., **additional form without interaction** provides the best fit to the data.
> We will revise the paper to include the corrected form, along with a derivation and ablation comparison of alternative models.
>
> # Comment on more experiments
> - We agree that additional experiments on compute optimality could further strengthen the claims in Section 5.2. However, this would require sweeping over many configurations with comparable FLOP budgets, which is not feasible under our current computational constraints. Such experiments would require multiple times the total GPU budget we already spent (estimated in Appendix A.10), amounting to over 10,000 USD even under conservative assumptions (e.g., 1 USD per GPU hour).
> - Moreover, while compute optimality is important, it is not the primary focus of this work. Our main goal is to understand how upcycling performance depends on data and model sizes. That said, our formulation naturally supports compute analysis, and we use it in Section 5.2 to identify regimes where upcycling becomes less compute-efficient than training from scratch. Note that our findings are robust and extrapolatable: in **Figure 6**, we show that scaling behavior observed in sub-0.5B parameter models can **reliably predict** the performance of larger models (up to 1B parameters).
> - Our approach is also in line with prior work, e.g., Krajewski et al. (2024) derived a data–model size scaling law and use it to predict FLOP-optimal behavior, rather than performing exhaustive FLOP-based sweeps, emphasizing interpretable scaling relationships under realistic compute budgets.
>
> We will further correct typos and figure presentation as suggested in the revised version.
>
> We hope these address your concerns. Please let us know if further clarification would be helpful.

---

### Decision · Program_Chairs · 2025-05-01

**Decision:**

Accept (poster)

**Comment:**

This paper studies scaling laws for upcycling dense models into MoE models. With extensive experiments, the paper present scaling laws between dataset size and model configuration. Based on the observations, the paper propose that from scratch training could be more efficient in a certain condition.

Overall, the reviewers acknowledge the soundness of the experimental design to find the empirical scaling laws. The reviewers also appreciate the insights from the empirical experiments. On the other hand, the reviewers questioned about weak theoretical support and limited generalizability with limited model sizes. The authors provided more detailed theoretical analyses in the rebuttal to address those issues. Also, the authors clarified the limitation of compute resource to further scale up the experiments.

All three reviewers gave weak accepts in the review process and confirmed the rating after the rebuttal phase. Therefore, I would give a weak accept as the paper provides meaningful insights to the community.